# ADAPTIVE VIDEO UNDERSTANDING AGENT: ENHANCING EFFICIENCY WITH DYNAMIC FRAME SAMPLING AND FEEDBACK-DRIVEN REASONING

## ABSTRACT

Understanding long-form video content presents significant challenges due to its temporal complexity and the substantial computational resources required. In this work, we propose an agent-based approach to enhance both the efficiency and effectiveness of long-form video understanding by utilizing large language models (LLMs) and their tool-harnessing ability. A key aspect of our method is query-adaptive frame sampling, which leverages the reasoning capabilities of LLMs to process only the most relevant frames in real-time, and addresses an important limitation of existing methods which typically involve sampling redundant or irrelevant frames. To enhance the reasoning abilities of our video-understanding agent, we leverage the self-reflective capabilities of LLMs to provide verbal reinforcement to the agent, which leads to improved performance while minimizing the number of frames accessed. We evaluate our method across several video understanding benchmarks and demonstrate that not only it enhances state-of-the-art performance but also improves efficiency by reducing the number of frames sampled.

## 1 INTRODUCTION

Recent advancements in video understanding have been significantly driven by end-to-end pretrained large transformer models, particularly those built upon large language models (LLMs) Liu et al. (2023; 2024), known as multimodal LLMs. Despite these advancements, comprehending long form videos remains a considerable challenge due to prohibitive computational costs and suboptimal performance Dao et al. (2022). Various approaches have been proposed to extend the temporal context of video transformers, including techniques such as masking, attention approximations, and parametric memory modules (e.g. Wu et al. (2022), Piergiovanni et al. (2024)). However, these methods often add complexity by necessitating specialized architectures and training paradigms Song et al. (2024).

Efficient video processing requires strategic selection of relevant frames from the total video sequence Gao et al. (2023b); Li et al. (2024). Traditionally, methods in this domain mostly rely on uniform sampling Zhang et al. (2023a); Song et al. (2024) or selective retrieval from a subset of sampled frames Fan et al. (2024); Wang et al. (2023b). While these techniques improve processing efficiency by reducing the number of frames, they often lack adaptability, leading to potential redundancy.

To address the above shortcomings, here we propose a novel approach that leverages LLMs as adaptive agents for video understanding tasks. Our method utilizes the advanced reasoning, planning, and tool-use capabilities of LLMs (Pallagani et al. (2023); Zhao et al. (2024b); Schick et al. (2024)) to enhance sampling efficiency while maintaining robust performance in video understanding tasks. Specifically, our approach leverages a LLM-based agent that dynamically determines which frames to sample based on the specific context and query. This method ensures that frame selection is both relevant and efficient, effectively mitigating the limitations of static sampling methods.

Our approach draws inspiration from research indicating that humans strategically allocate attention and filter out irrelevant details based on the task at hand Lang et al. (2013); Heim & Keil (2012); Raymond et al. (1992). For example, when asked "*What is the main goal of the camera wearer in this video?*" versus "*What is the color of the bird that appears at the beginning?*", humans deploy distinct strategies: the former may necessitate a review of the entire video to understand its context, whereas the latter would involve focusing solely on the video's initial segment to identify the bird's color.

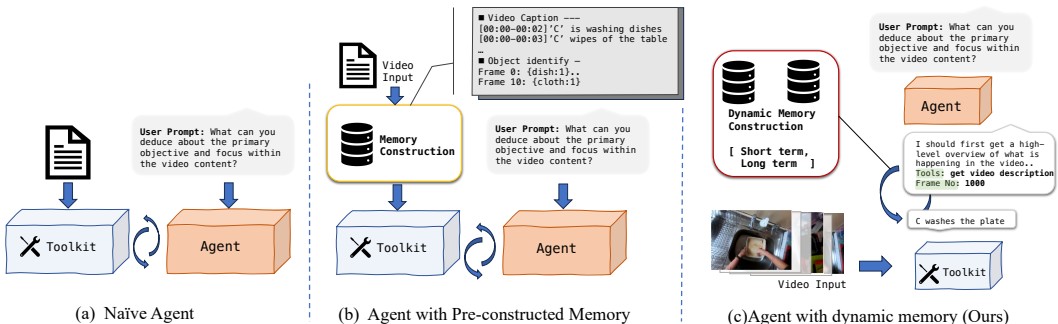

Figure 1: Comparison of methods: Our proposed method (c) is **query adaptive**, dynamically selecting frames based on query and video input to construct a responsive memory. In contrast, previous methods, including (a) Naïve agents and (b) Agents with pre-constructed memory, do not adapt to specific queries or utilize memory dynamically. We demonstrate that dynamically sampling frames have advantage over different set of benchmarks.

Our proposed framework adaptively samples and processes video frames in response to specific queries (see Figure 1 c). While previous approaches rely on static process which is independent of the query in extracting information such as captions Fan et al. (2024); Wang et al. (2023b), our approach attends to the given query and reasons strategically which frames to process during inference time without having to go through whole set of frames.

Our findings indicate that LLM agents, when used without guidance, exhibit suboptimal reasoning performance in terms of selecting the most informative frames. To enhance the reasoning ability of LLMs, we leverage the self-reflective capabilities of LLMs to provide insightful feedback Shinn et al. (2024); Pan et al. (2023a). Specifically, reflective statements serve as a form of verbal reinforcement, enabling the agents to develop an updated policy that facilitates more nuanced and sophisticated reasoning. Furthermore, our framework integrates long-term memory to store and utilize past experiences. The reasoning trajectories and the refinement is stored in the memory per instance. The key rationale behind adopting the memory is that retrieving past experiences that are relevant and semantically similar to a given query can significantly enhance the reasoning behavior of the LLM.

We validate the generalizability of our framework by evaluating it across a range of benchmarks, demonstrating its effectiveness and adaptability in various video understanding tasks. The results indicate that the proposed method outperforms existing approaches, achieving higher accuracy while maintaining a lowest number of frames accessed.

## 2 RELATED WORK

### 2.1 LONG CONTEXT MULTIMODAL AGENTS

Several approaches have been developed to handle multimodal inputs through agent-based reasoning Gao et al. (2023a); Yang et al. (2024); Fan et al. (2024); Wang et al. (2023b). These methods leverage agents' reasoning abilities along with their tool-calling capabilities. For instance, Yang et al. (2024) employs Monte Carlo Tree Search for reasoning combined with tool-calling techniques, while Gao et al. (2023a) utilizes ReAct Yao et al. (2022) for flexible video input processing.

Recent advancements have also focused on long-context videos Fan et al. (2024); Wang et al. (2023b). For example, Fan et al. (2024) uses memory retrieval during inference to address specific queries, which can be effective for localizing detailed information but may become redundant depending on the query type. Similarly, Wang et al. (2023b) relies on predefined sampling methods, necessitating extensive frame access for caption generation. Wang et al. (2024a) aims to reduce frame access by using a predefined number of frames and dynamic sampling, but primarily addresses short-form videos and straightforward question-answering scenarios.

Existing methods for addressing long-context processing using agent based approach (see Fig 1, b) involves preprocessing and extracting relevant information from frames during a pre-processing stage, with the agent retrieving memory dynamically based on the question during runtime Fan et al. (2024); Wang et al. (2023b). Although this approach can be effective, it is resource-intensive in terms of

memory and processing time. Additionally, it operates in a static manner, irrespective of the specific question, which can be redundant.

## 2.2 FRAME SAMPLING METHODS

Several methods have been proposed to enhance the efficiency of video frame handling by selectively subsampling relevant frames based on the content of the question or text, rather than using uniform sampling Gao et al. (2023b); Li et al. (2024); Yu et al. (2024); Pan et al. (2023a). For example, Romero & Solorio (2024) use CLIP model to retrieve pertinent frames through text prompts, while Han et al. (2023) propose a sampling technique that selects the most significant frames based on learned patterns. Although these approaches are effective, they often require pre-defined number of frames to sample or accessing to near all video frames to identify the relevant ones. These static ways of sampling frames may induce inefficiency as the video length becomes longer with exhaustive number of frames.

In contrast, our approach is inspired by human cognitive processes, which adaptively focus on information pertinent to the task at hand Lang et al. (2013); Heim & Keil (2012); Raymond et al. (1992); Heim & Keil (2017). We propose an agent that reasons about which frames to select based on the information from the question or previously extracted information, thereby improving the efficiency of information processing. While our method is similar to Wang et al. (2024b) in its query-adaptive nature, our method avoids the need for preprocessing (e.g., KNN clustering), thereby mitigating time-consuming operations.

| Model | Long-Context | Query Adaptive Sampling | Long-term Memory | Reasoning |
|---|---|---|---|---|
| AssistGPT Gao et al. (2023a) | ✗ | ✗ | ✗ | ReAct |
| DoraemonGPT Yang et al. (2024) | ✗ | ✗ | ✗ | MCTS |
| VideoAgent Fan et al. (2024) | ✓ | ✗ | ✗ | ReAct |
| LifelongMemory Wang et al. (2023b) | ✓ | ✗ | ✗ | Prediction Ensemble |
| Ours | ✓ | ✓ | ✓ | Refinement + ReAct |

Table 1: Comparison of existing methods. Previous approaches attempted to handle long-form video agents, however, our approach focuses on addressing long-context videos, adopting query adaptive sampling and long-term memory.

## 3 ADAPTIVE VIDEO UNDERSTANDING AGENT

We propose **A**daptive **V**ideo **U**nderstanding **A**gent, which reasons about which frames to process based on the observations and interactions made between the tools. Inspired by recent advancements in self-reflective ability of LLMs Jang (2023); Pan et al. (2023b); Shinn et al. (2024), we utilize the error feedback of LLMs to enhance the reasoning of the agent. We formulate the task likewise: The dataset $\mathcal{D} = (Q, A, V)$ consists of question $Q$, answer $A$, and corresponding $V$. The agent $\mathcal{L}$ is equipped with available actions $\mathcal{A}$. The agent $\mathcal{L}$ has only access to the meta-data of the video $V'$ (e.g. the total number of frames).

**Generating Policy** As illustrated in Figure 2, the initial step involves generating a policy $\pi$ based on the question and the details of the video. This policy encompasses an analysis of the question type and a detailed question analysis, which includes a sampling strategy and identification of key elements that the agent should focus on during the reasoning process. The policy serves a dual purpose: it guides the agent in planning and reasoning, and it can be abstracted and utilized in long-term memory. The rationale behind this approach is that, while the specifics of the question may vary, the abstracted high-level question type can be retained and leveraged in a manner similar to how humans utilize their generalized experiences.

**Planning/tool invoking** At time step $t$, the agent $\mathcal{L}$ selects an action $a_t$ and action input $x_t$ based on policy $\pi$ in solving problem $\mathcal{D}$. The actions $\mathcal{A}$ are the invokable tools, which are pre-defined and callable functions from the agent. The action input $x_t$ is typically the frame number, indicating which frames the tools should access. The input often includes extra arguments, for example the question to query the tools (e.g. Frame index 0, what is happening in the frame?). Once the tools are invoked, it returns a observation $\mathcal{O}$ which is the extracted information of the selected frame. The agent $\mathcal{L}$ considers the previous observation-action trajectory $\tau_t = [a_1, o_1, \ldots, o_{t-1}]$ : in choosing

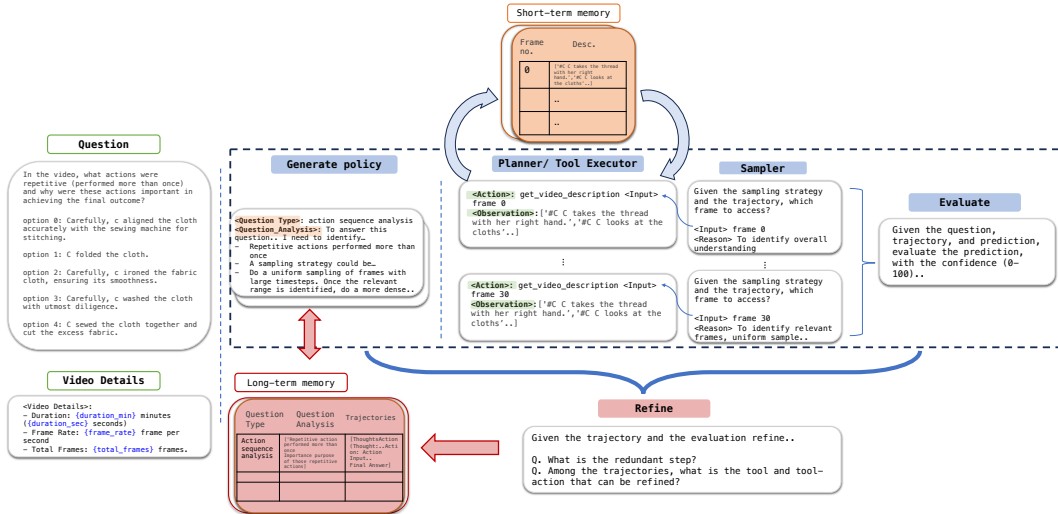

Figure 2: **Overall Framework**. The video metadata and question are provided to the agent to **generate policy**, which includes analyzing the question type and determining the task-solving strategy, including the sampling strategy. **Planner/Tool Executor**, based on the ReAct-style reasoning, generates thought processes, actions, and action inputs, and receives observations from the tools. During this stage, the sampler may suggest improved frames. After formulating the final answer, the evaluator and refiner are applied. The final result is then stored in **long-term memory**.

which actions to call.

$$a_t = \mathcal{L}(\pi, \mathcal{D}, \tau_{t-1})$$

Specifically, the agent $\mathcal{L}$ navigates search space, $\mathcal{F} \times \mathcal{A}$, where $\mathcal{F}$ represents the set of frames within $\mathcal{V}(|V| = |F| = n)$. The main goal of the agent $\mathcal{L}$ is to effectively prune the search space (i.e., minimize the number of the frames access) while ensuring performance (i.e., maximizing the reward $r$). While making a decision of which action $a_t$ to take along with the action inputs, the agent collaborates with the **Sampler**, another instantiated LLM, which is responsible for suggesting which frames to select. The sampler suggestions are based on the previous action-observation trajectory.

**Evaluator** We introduce an evaluator $\mathcal{E}$, which assesses the correctness of the prediction based on the question and the trajectory. It employs an error-feedback mechanism, iterating through trial-and-error to identify model errors. The evaluator $\mathcal{E}$ receives the question $\mathcal{Q}_i$, policy $\pi_i$ and the trajectory $\mathcal{T}_i$ and makes an judgment whether the final answer made by the planner is valid or not. The evaluation is made in a binary style True or False with a confidence ranging from 0 to 100.

**Refiner** Once the evaluation is done, the refiner is given a question, policy, and the trajectory from the agent, and the evaluation to generate the refinement of the trajectory. Specifically, the refiner first generates diagnosis of the trajectory (e.g., if there is any redundant steps, or any actions or action input that can be refined). Then, it generates a refined plan. The refinement is generated regardless of the evaluation result. The reason behind this is that if the evaluation is correct, the refinement is stored along with the trajectory in the long-term memory to enhance the reasoning of future trials and if the evaluation if false, the refinement have direct purpose of refining the reasoning of the agent for the next trial.

**Long/Short Memory** We maintain the memory with Long-term memory $\mathcal{M}_{\text{long}}$ to store experiences, short-term memory $\mathcal{M}_{\text{short}}$ to store accessed frame information. This format allows us to utilize the long-term memory. When the The long-term memory $\mathcal{M}_{\text{long}}$ is present, it is indexed by the question type based on their semantic similarity, retrieving the semantically similar experiences (question type, and the trajectories). We append those memory in-context as a chat-history to let the agent be aware of previous experiences.

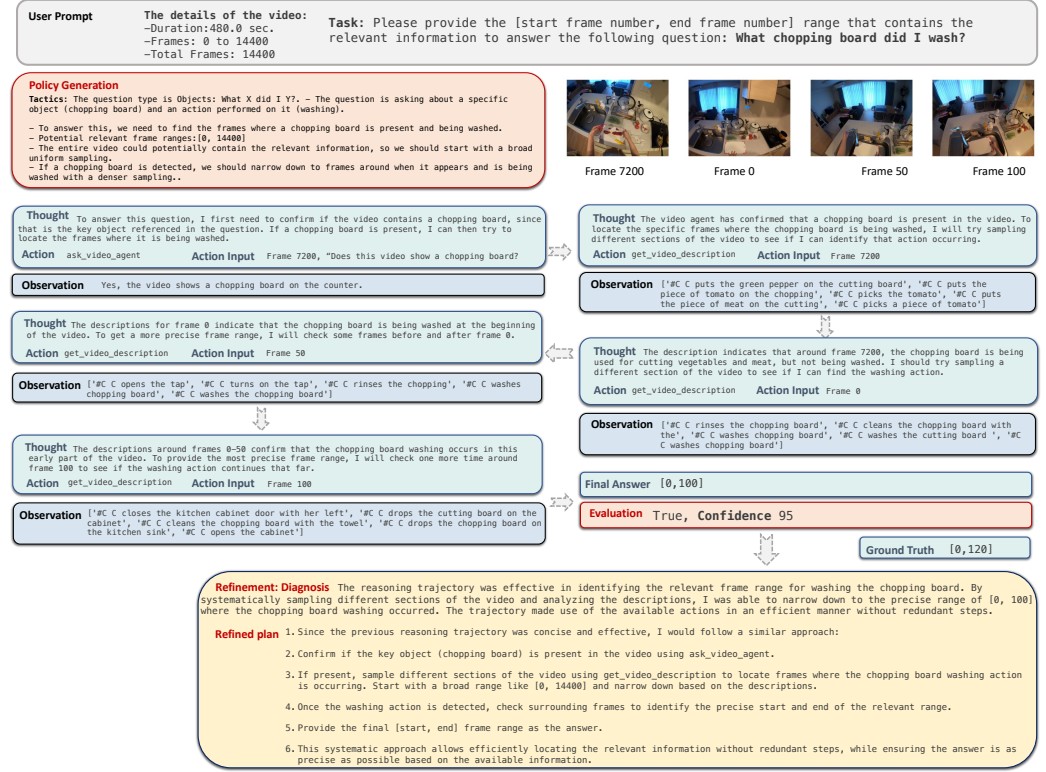

Figure 3: **Example of Ego4d NLQ Instance**. The User Prompt includes the video's metadata and the question for the Agent to address. (1) Policy Generation: the agent generates an analysis of the question and a sampling strategy (2) Thoughts, Actions and Observation: The agent formulates a Thought based on current state, executes an Action $\mathcal{A}$, with Action Input, and uses tools to obtain an Observation $\mathcal{O}$. This process iterates until the agent comes up with the final answer. (3) Evaluation: the Final Answer is assessed. (4) Refinement: The trajectory $\mathcal{T}$ is refined, and the results are stored in Long-term Memory $\mathcal{M}_{\text{Long}}$.

| Task | Source | Function |
|------|--------|----------|
| Video Caption Generation | LaViLa Zhao et al. (2023) | Detect actions, and objects |
| Video QA | Video-LlaVa Lin et al. (2023) | Extract Information |
| Image QA | Claude 3 Sonnet Anthropic (2024) | Image description |
| Object Tracking | RT-DETR Zhao et al. (2024a) ByteTrack Zhang et al. (2022) | Object detection |
| Text Caption | PaddleOCR PaddleOCR (2024) | Text caption Capture |
| Audio Transcription | Whisper Radford et al. (2023) | Audio capturing |

Table 2: **List of Invokable Tools**. This includes multi-modal tools, video-based tools (e.g., LaViLa, Video-LLaVa), image-based tools (e.g., Claude-3-Sonnet, PaddleOCR), and audio-based tools (e.g., Whisper)

## 4 EXPERIMENTS

### 4.1 TOOLS

In the experiments, the LLM used for reasoning and tool invocation is `Claude-3-Sonnet` Anthropic (2024). The tools used in the framework are detailed in Table 2. The tools are chosen to support multi-modalities, such as video, image, or audio. The **Video Caption Generation** model, LaViLa Zhao et al. (2023), generates descriptions for selected frames. To accommodate the model's requirement for frame sequences, we sample 3 additional frames (for a total of 4) for information extraction. Similarly, the **VideoQA** model, Video-LlaVa Lin et al. (2023), samples 3 additional frames (totaling 4) for video frame analysis. The **Object Tracking** model, RT-DETR Zhao et al. (2024a), identifies

objects with a confidence level above 0.6. The text caption tool PaddleOCR (2024) outputs text only if it is present in the frame.

## 4.2 EVALUATION DATASETS

| Dataset | Task | Example | Avg duration | # Instances |
|---------|------|---------|--------------|-------------|
| Egoschema | Action and scene understanding, abstract reasoning | *Q: What is the overarching behavior of C and the man in the video?*
*Option 0: C teaches the man game rules but the man seems distracted and is not paying attention*
*. . . .*
*Option 5: The man shows C a new card game while C takes notes for future references*
*A: Option 3* | 3mins | 0.5k |
| Ego4d NLQ | Temporal Localization | *Q: "What did I pick up before leaving the party?"*
*A: [3410,4000]* | 8.7mins | 3.9k |
| MovieChat | Long-term video understanding | *Q: "When does the things in the video happens, ancient age, modern age or future?"*
*A: "modern age"* | 9.4mins | 0.5k |
| NextQA | Causal and temporal action Reasoning | *Q:"Why was the toddler in red crying at the end of the video?"*
*A: Fell backwards* | 44secs | 8.5k |

Table 3: Overview of the evaluation Datasets.These benchmarks evaluate video understanding through a video question answering format, focusing on specific focus (denoted as Task).The average video duration varies from short form (<1min) to long form (<10min).

**EgoSchema** Mangalam et al. (2024) comprises broad spectrum videos of daily human activities, three-minute egocentric video segments. Each question is associated with five possible answers, in multiple choice question answering format. To correctly answer the question, it requires long-term temporal understanding. In this paper, we use a subset of the Egoschema dataset, consisting of 500 question and answer pairs.

**Ego4D NLQ** Grauman et al. (2022) consists of egocentric videos capturing a diverse range of daily activities from individuals wearing cameras. The primary task involves temporal localizing relevant frames within these extensive video contexts (e.g. Where did I put X?). The task can be formalized, given a video $\mathcal{V}$ and a natural language question $\mathcal{Q}$, the goal is to identify a relevant frame window $A$, such that the answer to $\mathcal{Q}$ can be deduced from $A$. We utilize the validation set for evaluation. The average length of the video is around 8.7 minutes and the expected prediction time window is around 9.3 seconds.

**MovieChat** Song et al. (2024) encompasses a range of categories , including documentary and detective films. The benchmark involves questions such as identifying common objects, temporal elements (e.g., day, night), and various scenes through open-ended questions and answers. The average duration of the videos is 9.4 minutes. For our evaluation, we utilize the test set (Global mode) of this benchmark. As it involves open-ended questions, we utilized *Cluade-3.5-sonnet* as an evaluator to evaluate whether the prediction matches with the ground truth answer. To be rigorous, we made the evaluator to generate the confidence of its judgment, counting only the instances with confidence over 80 (out of 100) as correct.

**NextQA** Xiao et al. (2021) is a benchmark designed to assess various aspects of video understanding, including causal action reasoning, temporal action reasoning, and common scene comprehension. Compared to other evaluation benchmarks used in this study, NextQA focuses on relatively short video clips, with an average duration of 44 seconds. While it does not align with the long form video question-answer evaluation criteria, we include this benchmark to demonstrate the generalizability of our framework across short-from videos. Also, NextQA benchmark consists of questions with 'textual cues', for example, *Why was the toddler in red crying at the **end** of the video?*, it allows us to investigate the adaptive behavior of the agents when presented with questions with textual cues and without textual cues.

| Model | # Frames | Accuracy |
|---|---|---|
| MultiModalLLM | | |
| FrozenBiLM | 90 | 26.9 |
| InternVid | 90 | 32.1 |
| ViT | | |
| ShortViViT | 32 | 49.6 |
| LongViViT | 256 | 56.8 |
| Agent | | |
| LLoVi | 180 | 57.6 |
| VideoAgent | 180 | 60.2 |
| LifelongMemory | 180 | 62.4 |
| Ours | 14.27 | 66.98 |
| Total Avg Frames | 5400 (30 fps) | |

Table 4: **Egoschema** Results. The number of frames accessed and Accuracy.

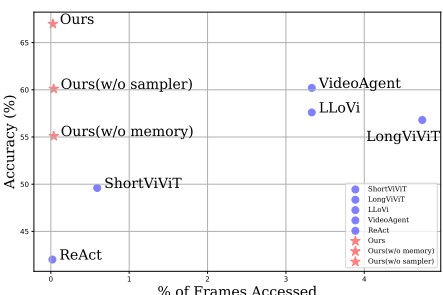

Figure 4: **Frames Accessed Ratio vs. Accuracy(%)** Our method demonstrate reduced % of frames accessed while maintaining high accuracy.

### 4.3 BASELINES

We experiments with several strong baselines which are comprised of multiModal LLMs incorporating the visual components along with the textual querys as inputs. FrozenBiLM Yang et al. (2022) learns cross modalities by training image projection layer. Similarly, InternVid Wang et al. (2023a) uses a image captioning model along with transformer based text embeddings to align the image and the text. These methods work on fixed and limited number of frames. Vision transformer (ViT) based methods are based on vision transformer utilizing joint space time attention. ShortViViT and LongViViT Papalampidi et al. (2024) harness input masking strategy, supporting prefixed number of frames 32 frames and 256 number of frames respectively.

We also experiment with agent-based methods, which utilize language models as agents harnessing external tools to solve video question and answering task. LLoVi Zhang et al. (2023a) extracts captions and LLM tackles the QA task based on the extracted captions. Analogously, LifelongMemory Wang et al. (2023b) process extracted captions and adopts voting by confidence strategy to conclude answers. VideoAgent Fan et al. (2024) harness multiple tools to process video. These methods typically sample frames with predefined fps rate (e.g. 1fps).

## 5 RESULTS

### 5.1 PERFORMANCE ANALYSIS

**Egoschema Results** Table 4 shows the results of evaluation on Egoschema benchmark. Our proposed method achieves accuracy of 66.98% which is more than 4% improvement over the best performing basline method, LifelongMemory (62.4%). For the baselines, we also observe a trade-off between the number of frames accessed and accuracy. For instance, Multimodal LLMs Yang et al. (2022) and Wang et al. (2023a) use a fixed sampling of 90 frames, but achieve relatively low accuracy ( 30%), whereas agent-based methods achieve significantly higher accuracy but sample twice as many frames. In contrast, our approach, which dynamically accesses relevant frames based on reasoning, reduces the number of frames accessed by approximately 93% while maintaining significant accuracy improvements. Existing methods typically use a uniform sampling strategy (1 frame per second), leading to a static number of frames. Our method avoids preprocessing all sub-sampled frames, thereby enhancing both accuracy and efficiency (Fig 4).

**Ego4d NLQ Results** We evaluate the intersection over union (IoU) at top-1 recall. (Table 5). Our method surpasses the baselines by 2% for IoU=0.3(%). Specifically, our method shows large improvement in IoU=0.5(%), which is around 10 % larger than the agent approach, and 11 % larger than the supervised approach. This may be attributed to the adaptive sampling strategy, which dynamically samples the frames, allowing both fine grained and coarse sampling. The frames are accessed on average 80% less than the agent method.

| | | IoU=0.3(%) r@1 | IoU=0.5(%) r@1 | #Frames |
|---|---|---|---|---|
| Supervised | 2D-TAN | 5.04 | 3.12 | 1024 |
| | VSLNet | 5.45 | 6.63 | 1461 |
| Agent | VideoAgent | 17.38 | 7.47 | avg 487(1fps) |
| | LifelongMemory | 15.99 | - | avg 487(1fps) |
| | **Ours** | 19.5 | **17.1** | avg 98 (0.002%) |

Table 5: Ego4d NLQ Results.

| | Accuracy | #Frames |
|---|---|---|
| VideoChat | 57.8 | 32 |
| VideoLlaMA | 51.7 | 32 |
| VideoChatGPT | 47.6 | 100 |
| MovieChat | 62.3 | 2048 |
| **Ours** | **84.8** | 13.59 (0.1%) |

Table 6: **MovieChat** Results.

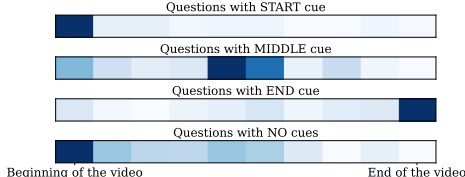

Figure 5: Frame accessed ratio based on textual cues from NextQA benchmark. Darker color corresponds to the higher ratio of access.

**MovieChat Results** Our method shows more than 22% increase in accuracy, while accessing only 0.1% of frames (Table 6), compared to the baseline models. This indicates that our method is more effective at processing long-form videos compared to both multimodal LLM–based (MovieChat Song et al. (2024)) and agent–based (VideoChatGPT, VideoLlama,VideoChat Maaz et al. (2023); Zhang et al. (2023b); Li et al. (2023)) baselines.

| | | Temporal | Causal | Descriptive | Average | # Frames (%) |
|---|---|---|---|---|---|---|
| Supervised | InternVid | 43.4 | 48 | 65.1 | 49.1 | 19.92 (1.8%) |
| | SeViLA | 61.3 | 61.5 | 75.6 | 63.6 | 39.85 (3.5%) |
| | MVU | 55.4 | 48.1 | 64.1 | 55.2 | 39.85 (3.5%) |
| Agent | LLoVi | 61 | 69.5 | 75.6 | 67.7 | 39.85 (3.5%) |
| | VideoAgent | 64.5 | 72.7 | 81.1 | 71.3 | 39.85 (3.5%) |
| | **Ours** | **71.42** | 69.1 | 77.77 | **72.7** | 12.37(1.1%) |

Table 7: **NextQA results** The NextQA results are categorized by question types: temporal or causal reasoning and descriptive QA. Our method achieves a +1.4% higher accuracy compared to baseline methods, while accessing 2.4% fewer frames.

**NextQA Results** Our method shows a 1.4% improvement in overall average accuracy (Table 7). When analyzed by question type—temporal, causal, and descriptive—our method particularly excels in temporal reasoning tasks, providing around 6.9% absolute improvement over the next best method.

## 5.2 ABLATION ANALYSIS

**Agents Without Guidance are Suboptimal Reasoners** LLM agents using the default ReAct reasoning, without any intervention, exhibit suboptimal performance (Table 8, 9 ReAct). This approach results in both low accuracy and a reduced percentage of frames accessed. Although LLMs have the potential to examine all avaiable frames and provide accurate answers, they often produce suboptimal results with fewer frames access. This is similar to observations where LLMs given one-shot questions demonstrate less rigorous reasoning compared to those using chain-of-thought or step-by-step interventions Wei et al. (2022); Yao et al. (2024). Our framework, akin to the chain-of-though method, enhances reasoning by incorporating internal interventions, leading to more accurate answers even if it requires accessing more frames.

**Questions including textual vs. non textual cues** Our proposed framework suggests that agents are query-adaptive, meaning they sample more efficiently when textual cues are present, as these cues guide their focus. For instance, a question like 'Why was the toddler crying at the end of the video?' will direct the agent to focus on the end of the video. The NextQA benchmark provides a natural testbed for evaluating whether agents leverage textual cues, as it includes both types of questions. Results indicate that the questions with textual cues result in an average of 10.56 frames accessed (.008%), compared to 12.26 (.01%) for questions without cues. Figure 8 presents a detailed analysis, showing that the ratio of frame accessed correlates with the presence of textual cues in the query. (e.g., a higher ratio of frames accessed at the beginning when 'Start' cues are included).

**Ablation of a component results in accuracy drop** A clear trend demonstrated across benchmarks (Table 8, 9) is that ablating any component consistently reduces accuracy. For Egoschema, the largest accuracy drop occurs when the evaluator is removed, while for Ego4D, the sampler's removal has the greatest impact. Although accuracy trends are clear, the effect on the number of frames accessed is less consistent. For example, ablating the sampler or refiner generally increases frame access, whereas in Ego4D, it decreases. This indicates that the role of components like the sampler and refiner may vary with benchmark characteristics. Ego4D benefits from extensive frame search, while Egoschema needs a holistic video understanding. Thus, these components help balance frame access and accuracy depending on the benchmark's requirements.

| | Egoschema | | Ego4d | | |
| Model | # Frames(%) | Accuracy | # Frames (%) | IoU=0.3(%) r@1 | IoU=0.5(%) r@1 |
|---|---|---|---|---|---|
| ReAct | 12.87 (.0024) | 42.02 | 23.987(.00) | 3.71 | 3.7 |
| Ours | 14.27 (.0026) | **66.98** | 98 (.002) | **19.51** | **17.07** |
| -w/o memory | 20.57 (.0038) | 55.1 | 90.04 (.0022) | 9.09 | 9.09 |
| -w/o evaluator | 15.69 (.003) | 50.1 | 40.0 (.001) | 5.41 | 4.69 |
| -w/o sampler | 19.77 (.0037) | 60.1 | 55.67(.002) | 5.01 | 5 |
| -w/o refiner | 20.46(0.003) | 53.2 | 65.33(.001) | 5.1 | 3.5 |

Table 8: **Ablation results on Frames Accessed and Accuracy**. The ReAct model, with no interventions, exhibits the lowest accuracy and frame access ratio. Ablations of different components reveal varying performance trends. The ablation results of Moviechat and NextQA can be found in Table 9

## 6    CONCLUSION

In this paper, we introduced a novel framework for video understanding that addresses the limitations of current methods by leveraging the daynamic reasoning capabilities of LLMs. While traditional approaches often rely on static or uniform frame sampling, which can be inefficient and redundant, our method enhances sampling efficiency by enabling the LLM based agent to adaptively select relevant frames based on specific queries. The results from extensive benchmarking validate the effectiveness and adaptability of our framework, showcasing its ability to handle diverse video understanding tasks more efficiently than traditional methods.

## 7    LIMITATIONS

While our method demonstrated effectiveness across several benchmark tasks, it is important to acknowledge its limitations. First, the performance of our approach is dependent on the capabilities of the tools it utilizes. Variations in tool performance can directly impact the overall effectiveness of the framework. Additionally, reliance on API calls introduces potential latency issues. This dependency on external APIs may affect the consistency and speed of the processing.

## 8    BROADER IMPACT

The proposed framework for video understanding presents several broader impacts with potential implications across various domains. By leveraging dynamic LLM-based agents for adaptive frame sampling, our approach offers a more efficient and effective solution to the challenges of long-form video comprehension. This advancement could significantly enhance applications in fields such as automated video content analysis, surveillance, and multimedia indexing, where processing large volumes of video data is essential.

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

APPENDIX

A.CASE STUDY

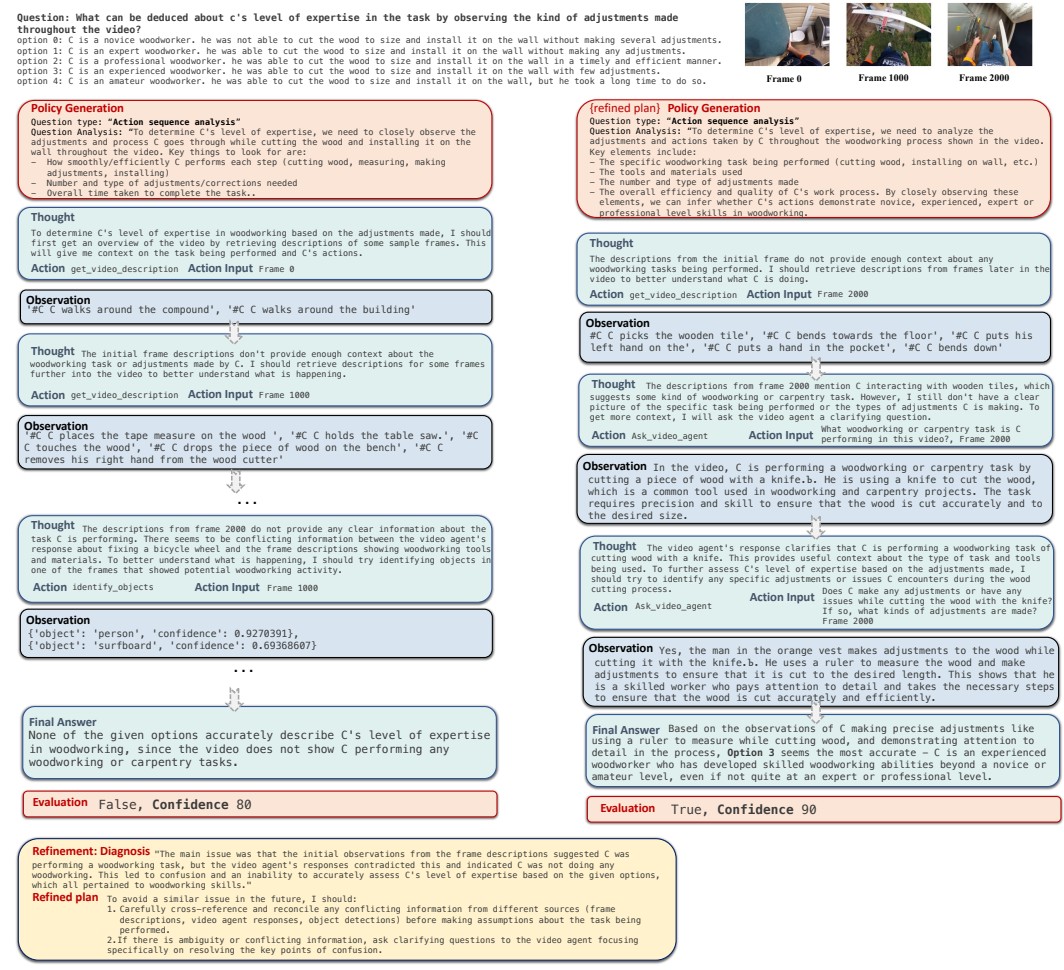

Figure 6: **Example of Egoschema Refinement**. Given the refinement based on the first trial (left), it attempts a second trial, with a refined policy $\pi^*$, which leads to the correct evaluation.

**Refined policy is more detailed and specific** What distinguishes the refined policy $\pi^*$ from the initial generated policy? Figure 7 illustrates examples of refined policies, where the initial trial produced an incorrect prediction, while the second trial yielded a correct one. Compared to the original policy, the refined policy is notably more detailed. Specifically, it includes: 1) updates in question analysis, and 2) a more nuanced approach to sampling strategies. Although the sampling strategies in both trials were largely similar, the refined policy offers a more granular description. For instance, while the initial policy merely referenced dense and sparse sampling, the updated policy provides a comprehensive overview of when to transition between different sampling strategies.

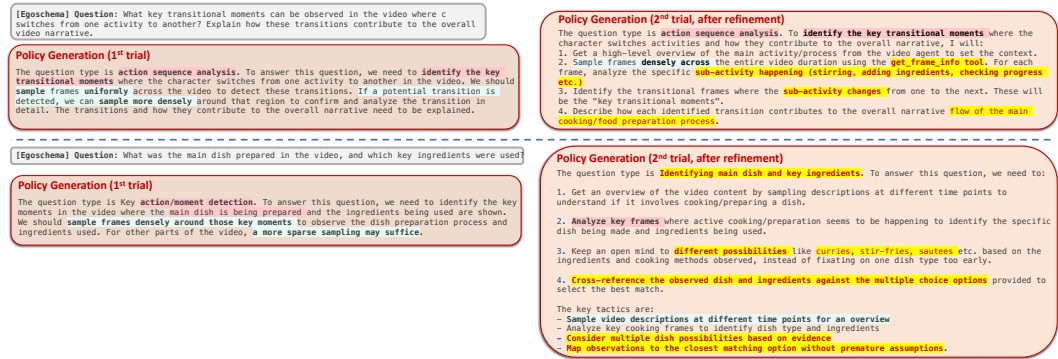

Figure 7: **Example of Refined Policy** Compared to the original policy, the refined policy is notably more detailed. Texts highlighted in yellow shows the added instruction.

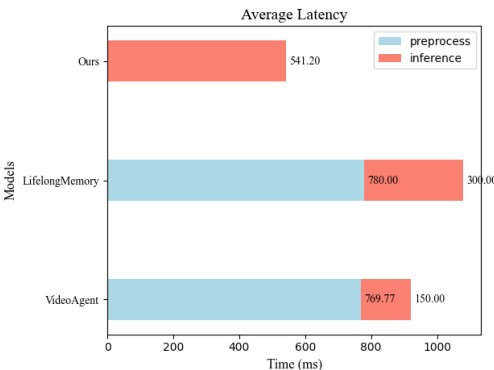

Figure 8: Latency Comparison with Other Agent Approaches: Our method reduces latency by processing videos only at runtime, compared to LifelongMemory Wang et al. (2023b) and Videoagent Fan et al. (2024), which require preprocessing.

## B. LATENCY ANALYSIS

## C. PROMPTS CONFIGURATION

### Policy Generation Prompt

```
  You are an advanced AI agent tasked with efficiently and accurately
processing video question and answering tasks.

You will be given a question related to a video, and you are responsibile
for coming up with a set of tactics and plans based on the characteristics
of each question.  The questions you will encounter will vary greatly,
ranging from inquires about the overall plot to specific details within
the video.

To effectively handle these tasks, you must first generate a set of tactics
and plans based on the characteristics of each question. You will be given
a question, please analyze the question.

- Determine the type of question (e.g. purpose/goal identification, tools
and materials usage, key action/moment detection, character interaction,
action sequence analysis..etc)

- How should the frames be sampled to solve the question? (e.g. Uniform
sampling with timestep 30.  If relevant frame is detected, more uniform
sample with timestep 2.)

{Question}
{Video details}
```

**Agent Prompt**

```
You are an advanced AI specialized in video question-answering tasks.
Your capabilities include executing necessary tools and interpreting their
outputs. Your objective is to select which frames to process and strategize
which tools to deploy and use their outputs to provide accurate answers to
questions related to a video.
<Video Details>:
- Duration: {duration_min} minutes ({duration_sec} seconds)
- Frame Rate: {frame_rate} frame per second
- Total Frames: {total_frames} frames.
- Frames with scene change: {scene_list}
Among the total {total_frames}, you will first choose sample frames to
understand the context. Please use the tool 'get_frame_info' to get the
general information of the frame. You can use the tools listed below. You
can reason what's happening between frames, and what's described in the
frame itself.
Use these tools to help: {tools}
Use the following format:
Thought: Consider what to do next.
Action: The action to take, using one of [{tool_names}].
Action Input: The input for the action.
You will receive the result of the action as Observation: The result of the
action. Please repeat the Thought/Action/Action Input/Observation cycle as
needed.
The final answer should be provided under 'Final Answer:' You must choose
one of the options among Option 0, Option 1, Option 2, Option 3, Option 4.
Please start with Thought:
Begin!
```

**Refiner Prompt**

```
You are an advanced reasoning agent that can improve based on self refection.
Your goal is to come up with a diagnoses and a refinement plan that is
effective in making a correct prediction.  You will be given a previous
reasoning trial in which you were given access to execute tools to solve
and an evaluation to the trial.
If the evaluation is False, you were unsuccessful in answering the question
either because you guessed the wrong answer with Final Answer, or you used
up your set number of reasoning steps. The optimal goal is to have concise
reasoning path without having redundant actions. Even if the evaluation is
True, you can improve the reasoning path by removing the redundant steps or
by refining the repetitive actions. In a few sentences, Diagnose a possible
reason for failure and devise a new, concise, reasoning paths that aims
to mitigate the same failure.  Be detailed as possible and use complete
sentences.
```

**Evaluator Prompt**

```
You are an advanced agent that evaluates whether the predicted answer
is correct or not. You will be given a question, reasoning trajectories,
and the final answer predicted by an agent. Please evaluate whether the
prediction is valid or not. You can give your confidence in percentage
(0-100). Remember that the reasoning and predictions are not always correct.
For example, Evaluation: True, Confidence: 90
```

Table 9: Ablation result of Moviechat and NextQA. The results are consistent in exhibiting a drop in accuracy when ablating any component. However, the trends of the number of frames accessed are not consistent, varying across the benchmark, and the components

|  | MovieChat | | NextQA | |
| --- | --- | --- | --- | --- |
| Model | # Frames | Accuracy | # Frames | Accuracy |
| ReAct | 10.62 | 69.4 | 9.87 | 47.27 |
| Ours | 13.59 | **84.8** | 12.37 | **71.6** |
| -w/o memory | 15.84 | 70.3 | 11.79 | 63.11 |
| -w/o evaluator | 15.31 | 72.43 | 13.59 | 58.97 |
| -w/o sampler | 15.51 | 80.2 | 13.87 | 60.2 |
| -w/o refiner | 15.51 | 70.2 | 11.8 | 65.42 |

```
Question: Where was the rim before I picked it?

Sampler Suggestions

> Start from the beginning of the video to identify the rim object and its initial
location before being picked up <Sampler Suggestion: frame_index 0>

> Perform uniform sampling with a large timestep (7200 frames ~ 1/2 of total 14402
frames) to quickly identify if the rim object is visible around the middle portion
of the video before being picked up <Sampler Suggestion: frame_index 7200>

> To identify the approximate range where the "rim" object appears before being
picked up, I will sample at frame 10800 which is around 3/4 through the video. If
the rim is not visible there, I can infer it likely appears in the final 1/4 of the
video before being picked up.<Sampler Suggestion: frame_index 10800>

> To identify the approximate range where the rim first appears, I will sample
frames closer to the start with a smaller timestep of 1800 (30 seconds at 60 fps).
If the rim is visible in this frame, the relevant range likely starts around here.
If not, I will continue sampling backwards from this point with an even smaller
timestep. <Sampler Suggestion: frame_index 1800>

…

> To find the first appearance of the "rim" object before it is picked up, I should
start watching the video sequentially from the very beginning (frame 0) since my
previous sampling attempts were unable to identify when the rim first shows up.
<Sampler Suggestion: frame_index 0>

> Since the rim is already visible in frame 0, I should sample a few frames ahead to
see if the rim's position changes or if it gets picked up in the next few frames.
Frame 30 seems like a reasonable next step to check for any movement or interaction
with the rim. <Sampler Suggestion: frame_index 30>
```

Figure 9: **Sampler Example** The Sampler examples demonstrates that it is able to 1) calculate the frames in terms of sparse sampling 2) Dynamically switch sampling fps, based on previous observation 3) Densly sample relevant frames

