# OpenReview forum: "Adaptive Video Understanding Agent:  Enhancing Efficiency with Dynamic Frame Sampling and Feedback-driven Reasoning"
_ICLR.cc/2025/Conference — Submitted to ICLR 2025_

### Official Review · Reviewer_fyRV · 2024-10-20

**Soundness:** 2
**Presentation:** 3
**Contribution:** 2
**Rating:** 3
**Confidence:** 4

**Summary:**

The paper proposes a new method that utilizes LLM as agents to tackle long video QA tasks. Compared to existing methods for this problem, the proposed method excels in invoking the tools on only a small number of frames. The proposed framework includes a policy generator, a planner, a frame sampler, an evaluator, a refiner, and a long-term memory module. The policy generator reads in the question and the video metadata, analyzes question type and content, and form a high-level plan for tool use. Based on the high-level plan, the planner chooses specific tools to invoke as well as the inputs. The frame sampler helps decide which frame to invoke the tool on. The evaluator assess the correctness of the prediction and provide a confidence score. The refiner analyzes the entire process and generates a refined high-level plan. The results of the refined trajectory is stored in the long-term memory for future reference. Experiments show that the proposed method outperforms existing methods on 4 popular evaluation datasets for long video QA (including Ego4D NLQ which is a temporal grounding benchmark but phrased as a QA task). Detailed ablation studies show that each component of the proposed framework is essential.

**Strengths:**

- Compared to existing methods, the proposed method invokes tool on significantly less frames.
- The proposed method enhances state-of-the-art performance on a suite of challenging long video QA benchmarks.
- The presentation of the paper is good, providing concrete examples on Ego4D about how the proposed framework answers the question through step-by-step reasoning and refinement. The authors also provides full prompts for different components of the model, allowing for easier reproduction of the results.

**Weaknesses:**

- The paper's main underlying assumption that invoking the tool on less frames means more efficiency is highly questionable. While the proposed method does invoke the tools on less frames, it requires much more complex reasoning from the planner and the sampler to decide which tool to use and the inputs to the tool every time. In my opinion, to more fairly access the cost of different methods, the authors should also consider other metric to compare, such as the total number of input and output tokens used in invoking the LLM API on a specific dataset.
- Some comparisons in the experiments might not be very fair. For example I believe both LifelongMemory and LLoVi only use the Video Caption Generation tool and do not have access to the other tools. While achieving SoTA on the reported benchmarks with the full system is remarkable, ensuring a fair comparison between the methods is also very important.
- Some results of the baseline methods cited by the paper are incorrect. For example, the EgoSchema result of LifelongMemory is reported on the full dataset (more challenging) than the subset the authors used. I would encourage the authors to check the latest versions of the cited papers for more accurate and up-to-date comparisons.

**Questions:**

- The idea to adaptively sample frames based on the query seems to be most beneficial when the number of questions asked for each video is very small. When we want to know about many details of the video, I think dense captioning is probably still needed. While I acknowledge that the long-term memory might help alleviate the problem since the policy generator can possibly reuse part of an existing trajectory if the current query is similar to a past query, I still think the proposed adaptive sampling method is not as general as clustering-based adaptive sampling methods (similar to the idea described in [1]). I would like to hear about the authors' thoughts on this.
- Is the example shown in figure 3 a real reasoning trajectory or just an illustrative, conceptual example? If it's a real reasoning trajectory, I wonder why the sampler chooses to sample frame 0 after sampling frame 7200 - it feels quite counterintuitive that it directly jumps from the middle of the video to the beginning. Also, at frame 7200, if the model just want to know whether a chopping board is present in the frame, why does it choose to use the ask_video_agent tool, rather than the image QA or object tracking tool.
- I think it would also be interesting to do more qualitative analysis on the model's behavior. What are the common failure modes? What are the frequencies of even tool being invoked? These questions are not as important as the points I mentioned earlier (including in the "weaknesses" section), but probably helpful to the community for future directions.

[1] Mohamed Afham et al. Revisiting Kernel Temporal Segmentation as an Adaptive Tokenizer for Long-form Video Understanding. ICCV 2023 Workshop.

---

> ### Author Response · Authors · 2024-11-25
>
> Dear Reviewer fyRV,
>
> Thank you for your thoughtful comments and feedback. We appreciate your insights and would like to clarify the points you raised.
>
> **Adaptively sampling benefits**: We acknowledge that dense captioning has its own advantages, as you pointed out. However, our focus in this work is on handling long-context videos, where the overhead of pre-processing and generating dense captions can be burdensome. In contrast, our approach emphasizes dynamically and adaptively sampling frames, which offers more flexibility and efficiency in processing long-context videos.
>
> **Figure 3 reasoning trajectory**: Regarding the reasoning trajectory in Figure 3, this is a real reasoning trajectory. It’s important to clarify that the agent’s reasoning process does not involve direct (human) guidance on which frames to choose or which tools to use. Instead, the selection of frames and tools is based on the agent’s internal behavioral analysis, informed by its reasoning process using the LLM. We hypothesize though that since the middle frame does not contain the answer, and neighboring frames often share high similarity, the LLM skips over the adjacent frames and instead focuses on the first frame.
>
> Thank you for your valuable feedback. We hope these clarifications address your concerns, and we welcome any further suggestions you may have.

---

> > ### Comment · Reviewer_fyRV · 2024-11-26
> >
> > I thank the authors for their response. However, many of my concerns and questions were not resolved. I will keep my ratings.

---

### Official Review · Reviewer_83ou · 2024-11-03

**Soundness:** 3
**Presentation:** 3
**Contribution:** 3
**Rating:** 5
**Confidence:** 5

**Summary:**

The authors proposed an agent-based framework to improve long-form video understanding. It consists of a planning module and invoking tools to sample video frames that are relevant to solve the video understanding task. They conducted experiments on video benchmarks including EgoSchema, MovieChat, Ego4D-NLQ, and NextQA to demonstrate the effectiveness and efficiency (access less frames) of this method.

**Strengths:**

-the proposed agent framework to approach long-form video understanding is novel and the setup is different from previous ones.

-the proposed method achieved good performance on a variety of video benchmarks.

-the proposed method is more efficient than previous models that it access less frames.

**Weaknesses:**

-does the toolkit needs to be pre-designed and coded? If so, how does the author respond to its limitation on the generalizability of this method?

-the method is not compared to sufficient baselines, especially on NextQA and EgoSchema.

-accessing less frames should in theory improve inference time, yet the agent framework setup might contribute oppositely. Is it possible to include an inference time performance with previous models on the benchmarks studied?

**Questions:**

See weaknesses.

---

> ### Author Response · Authors · 2024-11-25
>
> Dear Reviewer 83ou,
>
> Thank you for your valuable comments and feedback. We appreciate your insights and the opportunity to clarify our work.
>
> **Toolkit pre-designed and coded**: In the scope of our study, we propose an agent that utilizes tools. While these tools are pre-designed and coded in the current version of the paper, they are modularized, meaning that they can be easily adapted or extended for future use. This modularity allows for flexibility in applying the tools to different contexts or research problems.
>
> **Baselines**: We understand your concern, and we acknowledge that more recent works have been published after our submission. In response, we plan to include additional, up-to-date baselines in our revised manuscript to provide a more comprehensive comparison.
>
> **Inference time comparison**: In Figure 8, we present a comparison of average latency across various agent-based approaches. While our method has a higher inference time latency compared to some baselines, it is important to note that when considering the total latency involved in solving a question, our method actually demonstrates a significant advantage. Specifically, our method achieves a latency of 541.20 ms, whereas LifelongMemory has a latency of 1080.00 ms and VideoAgent 919.77 ms.
> Once again, thank you for your constructive feedback. We hope these clarifications address your concerns, and we look forward to any further suggestions you may have.

---

> > ### Comment · Reviewer_83ou · 2024-11-26
> >
> > Thank you for your feedback. My concerns are partially addressed.

---

### Official Review · Reviewer_ENfV · 2024-11-03

**Soundness:** 2
**Presentation:** 2
**Contribution:** 2
**Rating:** 3
**Confidence:** 5

**Summary:**

The paper proposes a new method to improve the efficiency and effectiveness of long-form video understanding. The method leverages large language models (LLMs) to dynamically sample relevant frames, based on the specific query and context, thus avoiding redundant or irrelevant frame processing. Additionally, the reasoning abilities of the video-understanding agent are enhanced through self-reflective verbal reinforcement, leading to improved performance while minimizing frame access.

**Strengths:**

The paper introduces a novel query-adaptive frame sampling method that dynamically selects relevant video frames based on the context and query, significantly reducing computational overhead.

**Weaknesses:**

1. **Lack of Code Release**: The paper does not provide the code or detailed instructions for reproducing the experiments, which hinders the verification and utilization of the proposed method by other researchers.
2. **Limited Comparison with Advanced Baselines**: The paper does not compare its method with more recent and advanced models such as Qwen2-VL[1], CogVLM2-Video[2], and InternVL2[3], which could provide a more comprehensive evaluation of its performance.
3. **Narrow Benchmarking**: The evaluations are limited to a few benchmarks. Including more comprehensive and diverse benchmarks such as MVBench[4], Video-MME[5], and LVBench[6] would provide a broader assessment of the method's capabilities.


### Reference:
- [1] Wang, Peng, et al. "Qwen2-vl: Enhancing vision-language model's perception of the world at any resolution." arXiv preprint arXiv:2409.12191 (2024).
- [2] Hong, Wenyi, et al. "Cogvlm2: Visual language models for image and video understanding." arXiv preprint arXiv:2408.16500 (2024).
- [3] https://huggingface.co/OpenGVLab/InternVL2-Llama3-76B#quick-start
- [4] Li, Kunchang, et al. "Mvbench: A comprehensive multi-modal video understanding benchmark." Proceedings of the IEEE/CVF Conference on Computer Vision and Pattern Recognition. 2024.
- [5] Fu, Chaoyou, et al. "Video-MME: The First-Ever Comprehensive Evaluation Benchmark of Multi-modal LLMs in Video Analysis." arXiv preprint arXiv:2405.21075 (2024).
- [6] Wang, Weihan, et al. "LVBench: An Extreme Long Video Understanding Benchmark." arXiv preprint arXiv:2406.08035 (2024).

**Questions:**

1. **Code Release and Reproducibility**: It is crucial to release the code and provide detailed instructions for reproducing the experiments. This will enable other researchers to validate and build upon your work.

2. **Comparison with Advanced Baselines**:
    - It would be beneficial to compare your method with more recent and advanced models like Qwen2-VL, CogVLM2-Video, and InternVL2. These comparisons could highlight the strengths and weaknesses of your approach relative to the state-of-the-art.

3. **Comprehensive Benchmarking**:
    - Include evaluations on more comprehensive and diverse benchmarks such as MVBench, Video-MME, and LVBench. These benchmarks offer a wider range of evaluation scenarios and could provide a more thorough assessment of your method's performance.

4. **Scalability and Latency**:
    - Provide more detailed discussions on the scalability and latency of your method. How does the proposed approach perform with increasing video length and complexity?

---

> ### Author Response · Authors · 2024-11-25
>
> Dear Reviewer ENfV,
>
> Thank you for your valuable feedback. We hope the following clarifications address your concerns.
>
> **Lack of Code Release**: We plan to release the code soon, and we appreciate your comments in this regard.
>
> **Advanced Baselines**: The choice of baselines in our study was driven by the need to evaluate methods specifically designed to address the challenges of long-context videos. While we aim to incorporate the most up-to-date relevant works, some of the papers you mentioned were published after our submission. However, in response to your suggestion, we plan to expand our experiments and include comparisons with more advanced baselines.
>
> **Narrow Benchmarking**: We selected the benchmarks in our study because they are specifically designed to handle long-context videos. While broader benchmarking could help assess generalizability, our primary focus is on methods that address the unique challenges of long-context scenarios. We believe this focus provides the most relevant insights for our research goals.
>
> **Scalability and Latency**: We acknowledge your concern regarding scalability and latency. We plan to include a more detailed, fine-grained analysis of these aspects in our revised manuscript.
> Once again, thank you for your thoughtful feedback. We hope these clarifications help, and we look forward to any further suggestions you may have.

---

### Official Review · Reviewer_bFVh · 2024-11-04

**Soundness:** 2
**Presentation:** 3
**Contribution:** 2
**Rating:** 3
**Confidence:** 5

**Summary:**

This paper presents an adaptive video understanding agent that performs query-adaptive sampling on long-form videos with the assistance of LLMs. The method leverages the self-reflective capabilities of LLMs to guide the agent in selecting relevant frames, utilizing necessary tools, and refining the reasoning trace to derive the final answer. The authors demonstrate the effectiveness of the proposed method on popular videoQA benchmarks.

**Strengths:**

The idea of utilizing a memory is quite interesting and makes sense for long-form video QA tasks

The ablation study is very useful.

**Weaknesses:**

I would like the authors to clearly articulate their contributions/novelty, given the extensive literature on long-form video QA and the availability of numerous papers applying similar strategies (e.g., reasoning, memory, tools). I found it difficult to distinguish this paper from many existing studies. For instance, if the query-adapter sampler is the novel component, it is not being clearly positioned wrt literature and articulated why this is better than others and works better.

Furthermore, the lack of extensive and fair comparison to all the relevant literature is missing.

**Questions:**

1.	It is challenging to understand the key novelty of the paper, especially if it lies in the query-adaptive sampler. Unfortunately, the authors do not discuss how the LLM performs this efficiently or what makes this approach unique.

2.	The use of tools, evaluators, refiners, or long/short memory is not novel, as several papers like VideoAgent, MMCTAgent, VideoTree, DoremonGPT , already explore these approaches.

3.	The authors do not compare their approach with very relevant works, including:

a.	Kumar, Somnath, et al. "MMCTAgent: Multi-modal Critical Thinking Agent Framework for Complex Visual Reasoning." arXiv preprint arXiv:2405.18358 (2024).

b.	Wang, Ziyang, et al. "VideoTree: Adaptive Tree-based Video Representation for LLM Reasoning on Long Videos." arXiv preprint arXiv:2405.19209 (2024).

c.	Wang, Xiaohan, et al. "Videoagent: Long-form video understanding with large language model as agent." arXiv preprint arXiv:2403.10517 (2024).

4.	The authors also do not provide a clear analysis of the frames utilized and the accuracy achieved compared to baseline models like VideoAgent, LifelongMemory, MMCTAgent, and VideoTree. For instance, this paper uses Claude, whereas other techniques use GPT models, making the comparison less fair. Given the differences in base models and capabilities, it is difficult to assess the true performance of the proposed system. It would be great if the authors could do fair comparison with same base models or provide detailed discussion of the impact of different system choices like base models, etc.

5.	While the integration of multiple tools and memory is beneficial, the novelty of this approach seems limited, as similar methods have been explored in prior work.

6.  While the authors discussed about VideoTree earlier in the paper, in Table 7, Video tree results are missing, which is better than the proposed approached. Please clarify.

---

> ### Author Response · Authors · 2024-11-25
>
> Dear Reviewer bFVh,
>
> Thank you for your time and effort in reviewing our manuscript and for providing such valuable feedback.
>
> **Novelty of the paper** - Query Adaptive Sampler
> We appreciate your comment regarding the use of tools, evaluators, and refiners in prior work. While these concepts have been explored before, our key contribution lies in the application of query-adaptive frame sampling, which leverages the reasoning ability of the agents. Our approach is particularly focused on improving efficiency and performance when handling long-context videos. As demonstrated in the results (Table 4, Figure 4), our method enhances efficiency by reducing the number of frames accessed, while simultaneously increasing the accuracy of tasks.
>
> **Adding Relevant Works**: Thank you for your suggestion. One reason we have not included certain works is that our primary comparison is with methods specifically designed to address the challenge of long-context videos. However, we recognize the importance of a broader context, and we plan to include results from the additional works you mentioned in our future benchmark comparisons.
>
> **Applying GPT models**: We acknowledge your concern that using different base models could lead to an unfair comparison. We intend to address this by including results based on GPT models in our future work, ensuring a more balanced and comprehensive evaluation.
> Once again, thank you for your thoughtful feedback. We hope these clarifications address your concerns, and we look forward to any further suggestions you may have.

---

> > ### Comment · Reviewer_bFVh · 2024-11-26
> >
> > I thank the authors for their response. While the response acknowledges the concerns I have raised, I do not think sufficient clarifications or details are provided to answer my queries, esp wrt comparison to relevant SOTA techniques. I will keep my score.

---

### Official Review · Reviewer_NMK8 · 2024-11-05

**Soundness:** 3
**Presentation:** 2
**Contribution:** 3
**Rating:** 8
**Confidence:** 4

**Summary:**

Manuscript propose several improvements over the prior VideoAgent (Fan et al., 2024) framework for long-form video understanding. The key contributions are: 1) bringing in a reflexion-alike self-debug method to fix the potential error in the initial tool-use plan produced by LLMs; 2) a "dynamic memory" where the feature and content extraction happens on-the-fly; 3) some different choice of tools, including a "sampler". Results on challenging egoschema, movieqa, nextqa and ego4d-nlq prove the effectiveness of the proposed method.

**Strengths:**

+Overall the manuscript is clear and well-written. The research topic (LLM agents for complex tasks) is of interest to the NeurIPS community.

+The proposed improvements over VideoAgent are technically sound and well-motivated. Bringing in reflexion is straightforward but indeed quite helpful to the performances, as the results indicate.

+The evaluation of the proposed method is comprehensive and impressive. The gain on challenging benchmarks over prior arts, ex. VideoAgent is ndeed significant. I like the metric on how many frames are needed to complete the tasks.

**Weaknesses:**

I don't have major concerns about this manscript. Well done! Here are some minor points that I hope the authors can help me with in a rebuttal:

-Compared to the "agent with pre-constructed memory", I agree that building a dynamic memory can be more flexible, but it can be less efficient in terms of response time. Does the author have any analysis or comparison on this?

-In the ablation study (table 8), how can it possible to have an ablative version of the proposed agent that only has "evaluator" or "refiner"? I think the evaluator is necessary for the refiner as they are both essential components of the reflexion framework.

-Can the authors elaborate on their LLM choice(claude-sonnet)?

-The font size of the text in almost all graphics can be too small.

**Questions:**

See weaknesses.

---

> ### Author Response · Authors · 2024-11-25
>
> Dear Reviewer NMK8,
>
> Thank you for your valuable feedback and your interest in our work. We appreciate the time you took to review our manuscript.
>
> Response time in inference: In Figure 8, we present a comparison of latency across different agent-based models, specifically those using “agents with pre-constructed memory”. While it is true that our method has a higher inference time latency compared to the baselines, it is important to note that the “agent with pre-constructed memory” baselines require a preprocessing phase. When considering the total latency involved in solving a question, our method actually demonstrates a significant advantage. Specifically, our method achieves a latency of 541.20 ms, compared to 1080.00 ms for LifelongMemory and 919.77 ms for VideoAgent.
>
> Ablation with only evaluator or refiner: Thank you for pointing this out! You are correct, “without evaluator” indicates the version where the evaluator and refiner aren’t present. We will clarify that the “w/o evaluator” ablation refers to the version where both the evaluator and the subsequent refiner process are omitted. We will make this distinction clearer in the main text.
>
> Choice of LLM - Claude sonnet: The selection of Claude Sonnet was based on its general capabilities. We plan to extend our experiments and include GPT in future work to provide a fair comparison with other methods in the field.
>
> Once again, thank you for your insightful comments. We hope these clarifications address your concerns.

---

> > ### Comment · Reviewer_NMK8 · 2024-11-26
> >
> > Thank you -- all my concerns have been addressed.

---

### Meta-Review · Area_Chair_udaZ · 2024-12-20

**Metareview:**

This paper proposes an agent framework for long-video understanding, which adaptively selects frames to process based on the query and callable tools on the input video. For a given query, the steps include generating the policy, tool execution (based on ReAct), sampler (for selecting frames), evaluation and refinement, and maintaining a long/short term memory. Claude-3-Sonnet is used for generating the plan and calling tools (including different pre-trained image and video models). Results are on Egoschema, Ego4D NLQ, MovieChat, and NextQA, shows improvements compared to some baselines. While reviewers appreciate the importance of the task the method is tackling, and the query-adaptive frame sampling part of the approach, concerns were raised on whether the comparisons in the paper are fair (given that baselines use much less tools, and different LLMs), lack of comparisons to other relevant video agents (e.g. those listed by reviewer bFVh and ENfV), and experimental design (e.g. even though less frames were called, but more complex tools are used on each frame, did the total cost decrease?). The reviewers have provided valuable feedback on this paper, and authors are encouraged to incorporate their suggestions for future submissions.

**Additional Comments On Reviewer Discussion:**

Reviewers raised concerns on the lack of comparisons to other video agent baselines, need for additional analysis on the frames selected, and suggested experiments that would help improve the paper. These concerns remain after the rebuttal period, and most reviewers agree to reject the paper in its current state.

---

### Decision · Program_Chairs · 2025-01-22

Reject